# Access to care solutions in healthcare for obstetric care in Africa: A systematic review

Anjni Joiner[1,2☯]*, Austin Lee[3‡], Phindile Chowa[4‡], Ramu Kharel[3‡],
Lekshmi Kumar[4‡], Nayara Malheiros Caruzzo[5‡], Thais Ramirez[2‡], Lindy Reynolds[6‡],
Francis Sakita[7‡], Lee Van Vleet[8‡], Megan von Isenburg[9‡], Anna Quay Yaffee[4‡],
Catherine Staton[1,2‡], Joao Ricardo Nickenig Vissoci[1,2‡]

1 Department of Surgery, Duke University School of Medicine, Durham, NC, United States of America,
2 Duke Global Health Institute, Durham, NC, United States of America, 3 Division of Global Emergency
Medicine, Department of Emergency Medicine, Brown University, Providence, RI, United States of America,
4 Department of Emergency Medicine, Emory University School of Medicine, Atlanta, GA, United States of
America, 5 Physical Education Department, State University of Maringá, Maringá, PR, United States of
America, 6 University of Alabama School of Public Health, Birmingham, AL, United States of America,
7 Kilimanjaro Christian Medical University College, Moshi, Kilimanjaro, Tanzania, 8 Durham County
Emergency Services, Durham, NC, United States of America, 9 Medical Center Library, Duke University
School of Medicine, Durham, North Carolina, United States of America

☯ These authors contributed equally to this work.
‡ These authors also contributed equally to this work
* anjni.joiner@duke.edu

## Abstract

### Background

Emergency Medical Services (EMS) systems exist to reduce death and disability from life-threatening medical emergencies. Less than 9% of the African population is serviced by an emergency medical services transportation system, and nearly two-thirds of African countries do not have any known EMS system in place. One of the leading reasons for EMS utilization in Africa is for obstetric emergencies. The purpose of this systematic review is to provide a qualitative description and summation of previously described interventions to improve access to care for patients with maternal obstetric emergencies in Africa with the intent of identifying interventions that can innovatively be translated to a broader emergency context.

### Methods

The protocol was registered in the PROSPERO database (International Prospective Register of Systematic Reviews) under the number CRD42018105371. We searched the following electronic databases for all abstracts up to 10/19/2020 in accordance to PRISMA guidelines: PubMed/MEDLINE, Embase, CINAHL, Scopus and African Index Medicus. Articles were included if they were focused on a specific mode of transportation or an access-to-care solution for hospital or outpatient clinic care in Africa for maternal or traumatic emergency conditions. Exclusion criteria included in-hospital solutions intended to address a lack of access. Reference and citation analyses were performed, and a data quality assessment was conducted. Data analysis was performed using a qualitative metasynthesis approach.

**Data Availability Statement:** All relevant data are within the paper and its Supporting Information files.

**Funding:** AJ acknowledges support from the Society for Academic Emergency Medicine

Foundation (SAEMF)/Global Emergency Medicine
Academy (GEMA) Research Pilot Grant, RF2019-
004. The funders had no role in the study design,
data collection, analysis, decision to publish or
preparation of the manuscript.

**Competing interests:** The authors have declared
that no competing interests exist.

## Findings

A total of 6,457 references were imported for screening and 1,757 duplicates were removed. Of the 4,700 studies that were screened against title and abstract, 4,485 studies were excluded. Finally, 215 studies were assessed for full-text eligibility and 152 studies were excluded. A final count of 63 studies were included in the systematic review. In the 63 studies that were included, there was representation from 20 countries in Africa. The three most common interventions included specific transportation solutions (n = 39), community engagement (n = 28) and education or training initiatives (n = 27). Over half of the studies included more than one category of intervention.

## Interpretation

Emergency care systems across Africa are understudied and interventions to improve access to care for obstetric emergencies provides important insight into existing solutions for other types of emergency conditions. Physical access to means of transportation, efforts to increase layperson knowledge and recognition of emergent conditions, and community engagement hold the most promise for future efforts at improving emergency access to care.

## Introduction

There is a lack of access to healthcare for a range of emergent conditions on the continent of Africa. Much of this impaired access can be traced back to a paucity of available or reliable transportation resources; less than 9% of the African population is serviced by an emergency medical services (EMS) transportation system, and nearly two-thirds of African countries do not have any known EMS system in place [1]. Many Africans must resort to using personal or public transportation in his or her time of greatest need [2–7]. This is further complicated when emergent conditions occur at inconvenient times of day or night, when commonly used means of transportation may be unavailable. Lack of transportation or access to care can lead to delays in care or an inability to receive care for emergent conditions [8,9]. This lack of access may also limit the ability to provide timely treatment for less-emergent or non-emergent continuing care for a range of health conditions.

Healthcare access to care and transportation solutions in low- and middle-income countries (LMICs) are defined as specific interventions that aid the ability of a patient in accessing a healthcare facility, either by provision of a transportation solution, social or financial intervention, or education-based training interventions. Access-to-care solutions, including the development of prehospital and EMS care systems in all socioeconomic settings, have demonstrated a reduction in mortality from time-dependent conditions [9–13]. Given the value of these interventions in reducing morbidity and mortality, the World Health Organization has called for a need to increase access to healthcare, specifically the availability of prehospital care systems in LMICs [14].

Although many EMS systems remain underdeveloped in Africa, obstetric conditions are among the top two causes of EMS transports in the continent [1]. A global focus on improving maternal mortality has resulted in numerous programs aimed at strengthening emergency obstetric care in LMICs. Many of these interventions follow the three-delay model, which delineates 3 primary delays that contribute to maternal deaths in LMICs: 1) delays in the

decision to seek care; 2) the time from when the decision to seek care is made to the time of arrival to the most appropriate facility; and 3) the delay in receiving care due to facility delays [15]. Applying lessons learned from addressing maternal mortality in these settings and translation of these concepts to a broader emergency care perspective has been previously proposed [16].

This systematic review attempts to perform a qualitative metasummary of known, evidence-based interventions to address healthcare access to care and transportation problems for patients with emergency obstetric conditions in Africa. Through summarizing the available literature on emergency obstetric interventions in this region, we aim to define the quality and scope of these interventions as well as to provide a foundation for the translation of these interventions to a more holistic emergency care lens.

## Methods

### Protocol and registration

This systematic review is reported in accordance with the Preferred Reporting Items for Systematic Review and Meta-Analyses (PRISMA) Statement S1 Table. and is registered in the PROSPERO database (International Prospective Register of Systematic Reviews) under the number CRD42018105371 [17].

### Eligibility criteria

Inclusion criteria included: 1) transportation or access to care solutions for prehospital or follow up care; 2) trauma or maternal and obstetric conditions; 3) interventions in African countries. Exclusion criteria were interfacility transportation solutions. Language was not excluded and all types of studies were included. All manuscripts, regardless of year of publication, were included.

### Information sources

Articles in reference to transportation for hospital or clinic access in an African country were searched in the following electronic databases: PubMed/MEDLINE, Embase, CINAHL, Scopus and African Index Medicus. No language limitations were placed on initial article selection. We performed citation trailing for all included articles by screening both the references and citing articles identified through Web of Science and Google Scholar. A qualified librarian with prior experience in systematic review methodology performed the searches and assisted in refining search criteria (MV).

### Literature search

The initial search strategy is outlined in S2 Table. All studies up to the date of 10/19/2020 were included. All results were blindly screened by two authors in Covidence™ for the initial search and Endnote for the citation trailing. Disagreements were adjudicated by a third author.

### Study selection

Eight reviewers evaluated titles and abstracts of the retrieved articles (AJ, AL, PC, AY, LV, LR, RK, LK). Each abstract was reviewed independently by at least two reviewers. Full-text articles of the selected abstracts were then independently evaluated by at least two reviewers. Any disagreement was resolved by a third reviewer's opinion (AJ).

### Risk of bias

Risk of bias was determined using study quality assessment tools from the National Heart, Blood and Lung Institute for controlled intervention, cohort, cross-sectional, and pre- and post- with no control group studies [18]. Quality assessment for qualitative studies was performed using the Joanna Briggs Critical Appraisal Checklist for Qualitative Research, which appraises 10 items specific to qualitative methodology through "yes", "no", "unclear" or "not applicable" responses [19]. Three authors (NC, TR, AJ) categorized risk of bias as either low risk or high risk based on the number of indicators with unclear specifications on the quality assessment. Those studies categorized as low risk of bias had 3 or less indicators with unclear specifications that were not directly related to the provision of raw data or analysis. Those studies with 3 or more items with unclear classifications or at least one low-quality indicator was classified as having a high risk of bias. Studies that were classified as descriptions of programs with no analysis were excluded from risk assessment.

### Data extraction

Two reviewers (NM and TR) independently conducted data extraction by reviewing each full text manuscript. Disagreements were solved by consensus and a third reviewer's opinion (AJ) was sought if the disagreement could not be resolved. Data collected included general characteristics of the studies, including type of study, year of publication, location of the study, type of intervention, and study outcome. Type of study was classified into one of the following categories: cross-sectional, cohort/case control (observational), controlled intervention, uncontrolled intervention, economic evaluation, qualitative, mixed methods, or intervention development. Intervention development studies were defined as those that included a formative evaluation or description of an intervention without a formal study component or outcome data; the description of the intervention was the key focus of these studies.

### Data analysis

We used both qualitative and descriptive approaches in analyzing the pooled data. A descriptive approach was used to provide an overview of the study characteristics whereas a qualitative approach provided an in-depth analysis of the results through grouping of interventions and outcomes. Given the significant heterogeneity amongst and between the studies and interventions performed, a meta-analysis of pooled quantitative data was not feasible.

## Results

### Study selection

A total of 6,457 citations from 5 databases (PubMed/MEDLINE, Embase, CINAHL, Scopus and African Index Medicus) were screened for eligibility (Fig 1). A total of 73 studies were identified for data extraction, however, given the paucity of trauma-related articles, we elected to include only obstetric and maternal health-related articles for the final descriptive analysis for a total of 63 studies.

### Studies characteristics

General characteristics for the included studies are found in Table 1. As demonstrated in Fig 2, a total of 20 different African countries were represented, with a range of 1 to 19 studies per country. Uganda had the largest number of studies at 19, followed by Zambia with 10, and Tanzania with 9. The remainder of the countries had 3 or fewer studies, with the majority only having 1 per country. The majority of studies were non-controlled interventional (22) or

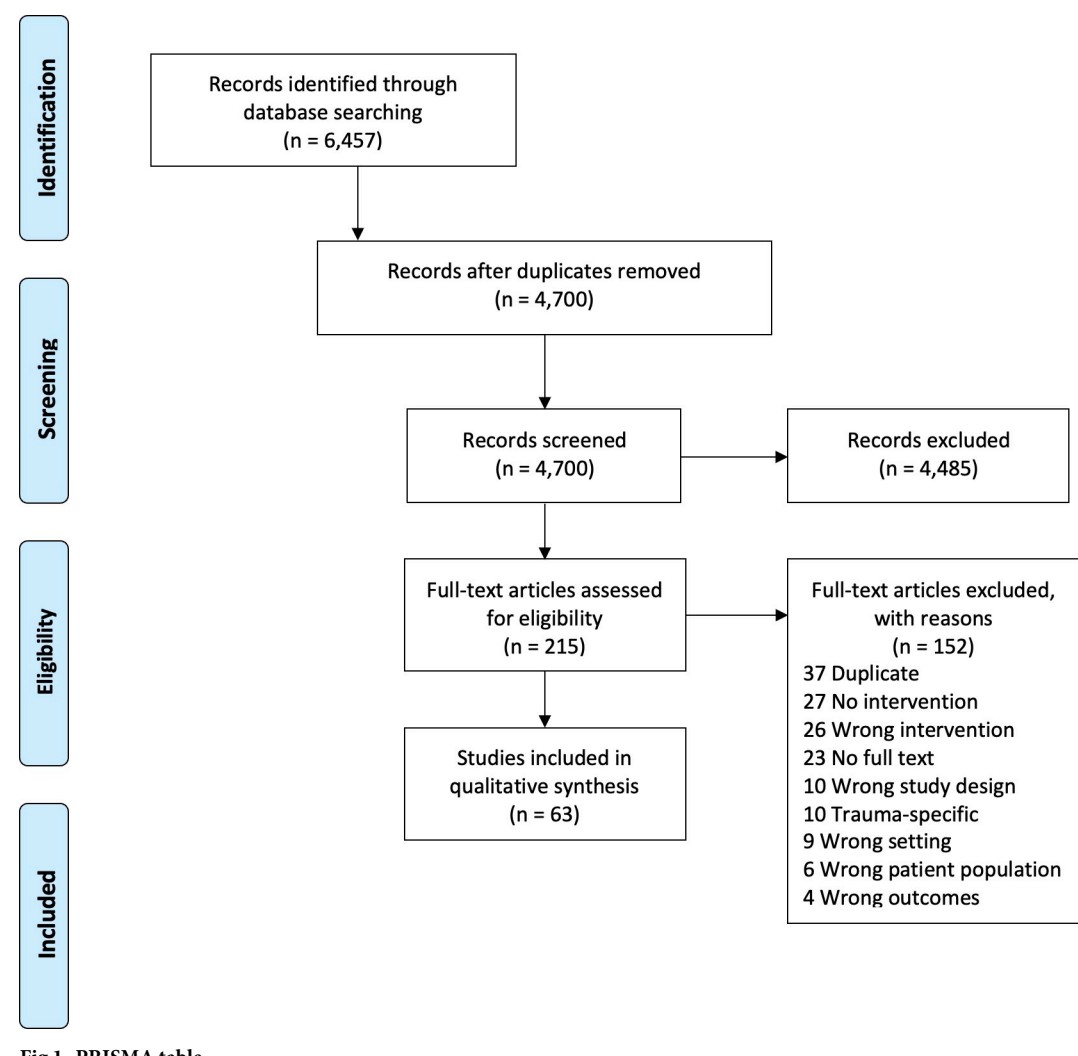

**Fig 1. PRISMA table.**

cross-sectional (10) studies. There were 8 mixed methods, 7 qualitative, 6 intervention development and 5 controlled intervention. The remaining 5 studies were 2 controlled intervention and 3 retrospective cohorts.

## Quality of studies

The vast majority of the studies (n = 51), were at a high risk of bias. Only 6 of the studies were at low risk of bias. Six of the included manuscripts were descriptions of interventions with no study performed and were therefore excluded from the quality assessment.

## Synthesis of results

The majority of studies [42] had multifaceted interventions involving more than one approach to improving access to care (Table 2). Of the 63 articles, we identified 28 that had at least one intervention in the category of community engagement, 27 with an intervention in the education/training category, 18 in financial category, 39 in the transportation category, and 16 in the facility infrastructure category. Of note, given the inclusion and exclusion criteria of this

**Table 1. Description of included studies by country, study design, area of focus, risk of bias and primary outcome.**

| Author, year | Country | Study Design | Area of focus | Risk of bias | Outcome |
|---|---|---|---|---|---|
| Accorsi, 2017 [20] | Ethiopia | Cross-sectional | Transportation | High | Cost-effectiveness, years of life saved |
| Adam, 2014 [21] | Kenya | Controlled intervention | Community Engagement | High | Knowledge about maternal and newborn health |
| Ahluwalia, 1999 [22] | Tanzania | Intervention development | Community Engagement, Training/Education, Transportation | n/a | n/a |
| Ahluwalia, 2003 [23] | Tanzania | Qualitative | Community Engagement, Training/Education | High | Community engagement and support |
| Ahluwalia, 2010 [24] | Tanzania | Mixed Methods | Community Engagement, Training/Education, Transportation | High | Sustainability and retention of Village Health Care Workers |
| Ahluwalia, 2011 [25] | Tanzania | Cross-sectional | Community Engagement, Transportation | High | Retention of community supported transport systems |
| Alfonso, 2015 [26] | Uganda | Non-controlled intervention | Financial, Transportation | High | Institutional delivery coverage rates, cost-effectiveness |
| Atnafu, 2016 [27] | Ethiopia | Cross-sectional | Training/Education, Transportation, Facility/Infrastructure | Low | Stillbirth rates, facility delivery rates |
| August, 2016 [28] | Tanzania | Controlled intervention | Community Engagement, Training/Education | High | Proportion of women who delivered in a healthcare facility; utilization of skilled care for antenatal care (ANC) visits; knowledge of danger signs during pregnancy, childbirth, postpartum, birth plan/complication readiness |
| Belaid, 2012 [29] | Burkina Faso | Qualitative | Training/Education, Financial, Transportation | High | Implementation challenges, perceptions |
| Bellows, 2013 [30] | Kenya | Non-controlled intervention | Financial | High | Odds of facility-based delivery |
| Bhopal, 2013 [31] | Sierra Leone | Mixed Methods | Transportation | High | Usage/acceptability, awareness of service, value of service, applicability of the concept to other areas |
| Cole-Ceesay, 2010 [32] | The Gambia | Intervention development | Community Engagement, Training/Education, Transportation, Facility Infrastructure | n/a | n/a |
| Conlon, 2019 [33] | Uganda, Zambia | Controlled intervention | Community Engagement, Training/Education, Transportation, Facility Infrastructure | High | Comparison of baseline and endline indicators of demand, timely access, quality, health systems strengthening, impact, and public and private healthcare facilities |
| Ekirapa-Kiracho, 2011 [34] | Uganda | Non-controlled intervention | Education/Training, Financial, Facility Infrastructure | High | Effectiveness of intervention, cost |
| Ekirapa-Kiracho, 2016 [35] | Uganda | Qualitative | Community Engagement, Financial, Transportation | High | Challenges, lessons learnt in supporting community efforts to improve maternal/newborn health |
| Ensor, 2014 [36] | Zambia | Non-controlled intervention | Community Engagement, Education/Training | High | Maternal health indicators (knowledge of antenatal care and obstetric danger signs, use of emergency transport, deliveries involving skilled birth attendants, etc.) |
| Essien, 1997 [37] | Nigeria | Cross-sectional | Financial, Transportation | High | Money raised, loans made, utilization of transportation, cost |
| Ezran, 2019 [38] | Madagascar | Cohort | Education/Training, Financial, Transportation, Facility Infrastructure | High | Antenatal care seeking (proportion of pregnant women seeking antenatal consultation at a public health facility) and perinatal care seeking (proportion of women who delivered at a public health facility) |

(*Continued*)

**Table 1.** (Continued)

| Author, year | Country | Study Design | Area of focus | Risk of bias | Outcome |
|---|---|---|---|---|---|
| Fiander, 2013 [39] | Tanzania | Non-controlled intervention | Financial | High | Number of referrals to CCBRT, referral regions and ambassador locations, transportation costs |
| Fofana, 1997 [40] | Sierra Leone | Non-controlled intervention | Community Engagement, Education/Training, Financial | High | Utilization of referral hospital by women with complications |
| Fournier, 2009 [41] | Mali | Non-controlled intervention | Education/Training, Financial, Transportation, Facility Infrastructure | Low | Number of women receiving emergency obstetric care, maternal mortality rates |
| George, 2017 [42] | Uganda | Intervention development | Financial, Transportation | n/a | n/a |
| Godefay, 2016 [43] | Ethiopia | Non-controlled intervention | Transportation | High | Maternal mortality rate |
| Grindle 2020 [44] | Uganda | Intervention Development | Financial, Transportation, Facility Infrastructure | n/a | n/a |
| Groppi, 2015 [45] | South Sudan | Cross-sectional | Transportation | High | Number of deliveries in hospital |
| Healey, 2019 [46] | Uganda, Zambia | Mixed Methods | Community Engagement, Education/Training, Transportation, Facility Infrastructure | High | Potential sustainability |
| Hofman, 2008 [47] | Malawi | Non-controlled intervention | Transportation | High | Referral time, costs |
| Jacobs, 2018 [48] | Zambia | Qualitative | Community Engagement, Education/Training | High | Process evaluation evaluating if the intervention was implemented as intended, any external factors that may have influenced the intervention, and factors contributing to gaps in achieved vs targeted outcomes |
| Jacobs, 2018 [49] | Zambia | Non-controlled intervention | Community Engagement, Education/Training | Low | Antenatal clinic attendance; antenatal care from a skilled birth attendant; receipt of intermittent preventive treatment during pregnancy; skilled birth attendant at delivery and at health facility; postnatal care within 48 hours after birth from an appropriate provider; and postnatal care within 48 hours from a safe motherhood action group (SMAG) |
| Kandeh, 1997 [50] | Sierra Leone | Cross-sectional | Community Engagement, Education/Training, Transportation, Facility Infrastructure | High | # patients seen at primary health units with complications, # patients referred by community motivators, cost |
| Kongnyuy, 2008 [51] | Malawi | Non-controlled intervention | Education/Training | High | Adherence to referral standards |
| Kruk, 2016 [52] | Uganda, Zambia | Controlled intervention | Community Engagement, Education/Training, Transportation, Facility Infrastructure | High | Obstetric provider knowledge, confidence, and satisfaction; patient experience of facility-based delivery at supported sites |
| Lalonde, 2003 [53] | Uganda | Non-controlled intervention | Education/Training, Transportation, Facility Infrastructure | High | Percentage of births in basic or comprehensive emergency obstetric facilities; case fatality rate among women with obstetric facilities |
| Magoma, 2013 [54] | Tanzania | Controlled intervention | Education/Training | High | Delivery in a health unit; postnatal care attendance, satisfaction |
| Massavon, 2017 [55] | Uganda | Controlled intervention | Financial | High | Service coverage of institutional deliveries, 4 antenatal care visits, and 1 postnatal care visit; cost per institutional delivery, cost per unit increment in institutional deliveries for each intervention |

(*Continued*)

**Table 1.** (Continued)

| Author, year | Country | Study Design | Area of focus | Risk of bias | Outcome |
|---|---|---|---|---|---|
| Mitchell, 2018 [56] | Senegal, Mali, Ghana, Nigeria, Ethiopia, Uganda, Kenya, Rwanda, Tanzania, Malawi | Non-controlled intervention | Community Engagement, Education/Training, Transportation, Financial, Facility Infrastructure | High | Multiple millennium development goal indicators and project-specific outcomes, including target attainment and on-site spending |
| Mucunguzi, 2014 [57] | Uganda | Non-controlled intervention | Transportation | High | C-section rate, hospital deliveries, average hospital stillbirths |
| Ngoma, 2019 [58] | Zambia, Uganda | Mixed methods | Community Engagement, Transportation, Facility Infrastructure | High | Facility delivery rates, description of strategies, methods of implementation, outputs, outcomes |
| Obare, 2014 [59] | Kenya | Non-controlled intervention | Financial | Low | Proportion of facility-based births |
| Oguntunde, 2018 [60] | Nigeria | Qualitative | Community Engagement, Education/Training | High | Comparison of stand-alone ETS initiatives with those integrated with demand creation activities |
| Onono, 2019 [61] | Kenya | Qualitative | Community Engagement, Transportation | High | Explore care seeking decision-making for pregnancy and childbirth services and perceptions of the intervention on the pregnancy and childbirth experience |
| Onono, 2019 [62] | Kenya | Cohort study | Community Engagement, Transportation | High | Adherence to recommended antenatal and postnatal care; skilled birth delivery |
| Pariyo, 2014 [63] | Uganda | Mixed methods | Financial | High | Number of institutional deliveries, costs, qualitative evaluation |
| Quam, 2019 [64] | Uganda, Zambia | Intervention development | Transportation, Facility Infrastructure | n/a | n/a |
| Rosato, 2012 [65] | Malawi | Mixed Methods | Community Engagement, Education/Training | High | Descriptive |
| Samai, 1997 [66] | Sierra Leone | Non-controlled intervention | Transportation | High | Referral hospital utilization, case fatality rate, cost |
| Santos, 2006 [67] | Mozambique | Cross-sectional | Transportation, Facility Infrastructure | High | Proportion of births in facilities, proportion of utilization among women with complications; case fatality rate |
| Satti, 2012 [68] | Lesotho | Non-controlled intervention | Community Engagement, Education/Training | High | Number of antenatal clinic visits and facility-based deliveries, number of patients with complicated deliveries transported to hospital |
| Schmid, 2001 [69] | Tanzania | Non-controlled intervention | Community Engagement, Transportation | High | Number of communities providing transportation support |
| Schmitz 2019 [70] | Uganda | Non-controlled intervention | Transportation, Facility Infrastructure | High | Estimated travel time to emergency obstetric and newborn care |
| Schoon, 2013 [71] | South Africa | Non-controlled intervention | Transportation | High | Maternal mortality, dispatch times |
| Serbanescu, 2019 [72] | Uganda and Zambia | Mixed Methods | Community Engagement, Financial | High | Maternal mortality |
| Serbanescu, 2019 [73] | Uganda and Zambia | Non-controlled intervention | Community Engagement, Financial | High | Emergency obstetric care indicators, institutional deliveries, obstetric complications, stillbirth and predischarge neo- natal mortality rates, maternal mortality |
| Sevene, 2020 [74] | Mozambique | Randomized-Control trial | Community Engagement, Education/Training, Transportation | Low | All-cause maternal, fetal, and newborn mortality and major morbidity |
| Sharma, 2017 [75] | Nigeria | Intervention development | Community Engagement, Education/Training, Transportation | n/a | n/a |
| Shehu, 1997 [76] | Nigeria | Mixed methods | Transportation | High | Number of obstetric complications transferred, costs |

(*Continued*)

**Table 1.** (Continued)

| Author, year | Country | Study Design | Area of focus | Risk of bias | Outcome |
|---|---|---|---|---|---|
| Sialubanje, 2017 [77] | Zambia | Qualitative | Community Engagement, Education/Training | Low | Perceived effectiveness of the program in increasing utilization of maternal health services |
| Somigliana, 2011 [78] | Uganda | Cross-sectional | Transportation | High | Cost-effectiveness |
| Tayler-Smith, 2013 [79] | Burundi | Retrospective Cohort | Transportation | High | Maternal mortality |
| Tayler-Smith, 2013 [80] | Burundi | Cross-sectional | Transportation | High | Cost, association between referral times and maternal/early neonatal deaths; coverage of complicated obstetric cases |
| Tsegaye, 2016 [81] | Ethiopia | Cross-sectional | Transportation | High | Number of ambulance referrals, number of complications and maternal deaths in referred patients, costs of ambulance services |
| Wilunda, 2016 [82] | Ethiopia | Non-controlled intervention | Community Engagement, Education/Training, Financial, Transportation, Facility Infrastructure | High | Attendance of at least 4 antenatal visits provided by a health professional or a health extension worker; receipt of all three basic services during antenatal care; delivery assisted by a skilled birth attendant; receipt of postnatal care visit within 7 days of delivery by a health professional or a health extension worker |

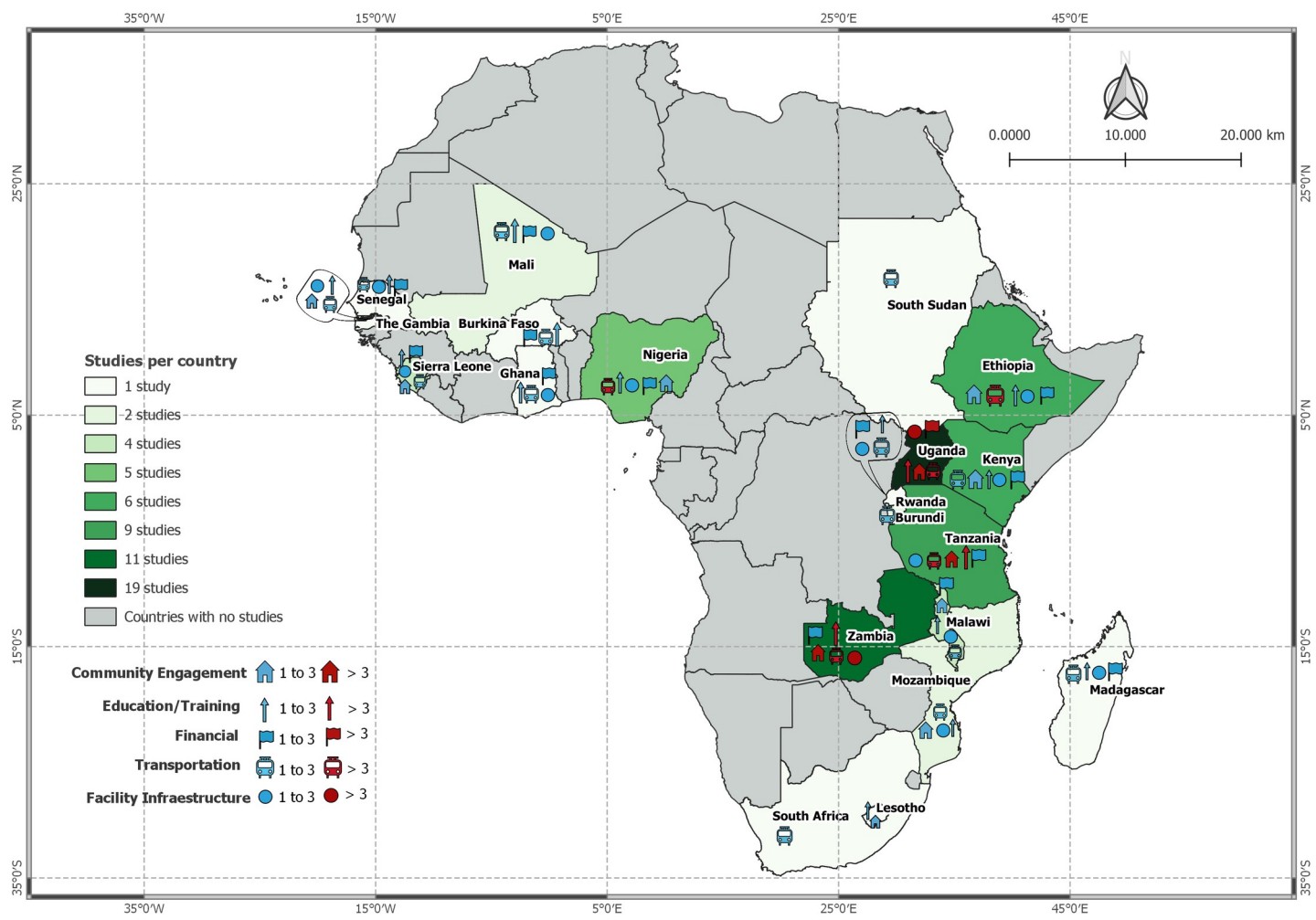

**Fig 2. Geographic distribution of included studies.**

**Table 2. Summary of intervention descriptions and outcomes, grouped by category.**

| Intervention type | Summary of outcomes |
|---|---|
| **Community Engagement** [21,23–25,28,33,35,36,40,46,48–50,52,56,58,60–62,68,69,72–75,77,82] | |
| Task shifting through volunteer community health workers (CHW) who were trained to provide health education and promotion of delivery with skilled birth attendants through health messages delivered to their respective villages [21] | Women exposed to CHW health messages had higher mean knowledge scores of maternal and newborn care and increased facility-based deliveries [21] |
| Community capacity building and empowerment initiative; integration of community village health workers (VHWs) and traditional birth attendants (TBAs); increasing participation by community members in planning and decision making [23] | Increased knowledge of danger signs, birth planning, timely referrals, and transport to hospitals; support and retention of VHWs [23] |
| Community capacity building and empowerment initiative; integration of community VHWs and TBAs; increasing participation by community members in planning and decision making [24] | Sustained VHW involvement in education, continued village emergency transport system use. Improved early prenatal care, identification of pregnancy danger signs, outcome data in targeted districts [24] |
| Community capacity building and empowerment initiative; development of community-supported emergency transport system via VHWs [25] | 6 out of 10 villages with community-supported transportation systems continued to function after completion of the initial intervention by integrating into the village infrastructure and governance [25] |
| CHWs were trained by health workers to provide education to community members through a program called Home Based Life Saving Skills which covered topics on maternal and newborn complications using cards depicting complications on one side and actions for treatment on the other side [28] | Increased facility-based deliveries after training, increased ANC visits, increased knowledge of danger signs and birth plan/complication readiness [28] |
| The Saving Mothers Giving Life (SMGL) program applied both supply- and demand-side interventions to address the 3 delays to maternity care through using district health systems by training CHWs, communicating through materials and mass media, transportation vouchers, improving infrastructure, development of maternity waiting homes, procurement of emergency transport vehicles, increasing facility access, training, quality improvement, and capacity building [33] | 44% reduction in both facility and districtwide maternal mortality ratio (MMR) in Uganda, and a 38% decrease in facility and a 41% decline in districtwide MMR in Zambia [33] |
| Increasing community awareness and engagement through community health team home visits, community dialogues, talk shows, and radio; supporting savings groups and improving linkage with local transporters [35] | Increased awareness of birth and neonatal health issues, improved access to transportation via linkage from savings groups [35] |
| Safe Motherhood Action Groups (SMAG) to discuss safe pregnancy, delivery, and neonatal care; encouraged communities to ensure transport and social support were available for trips to healthcare facilities [36] | Improved women's knowledge of antenatal care and obstetric danger signs; increased use of emergency transport; increased deliveries involving a skilled birth attendant [36] |
| Community health education program; community-generated loan funds [40] | Increase in use of referral facility by women from areas with community loans; cost of community immobilization was $472 U.S. dollars [40] |
| SMGL program: Community engagement-specific interventions included training and engagement of community groups and leaders; SMAGs in some districts [46] | Community volunteer groups were formalized and institutionalized; continued leadership promoting maternal health; increased maternity waiting homes; cost effectiveness via cost of death averted between $177–206 per life year gained [46] |
| SMAGs were implemented to increase awareness of maternal and neonatal health services and to link the community with available services, including ANC, postnatal care (PNC) and skilled birth attendant (SBA) visits [48] | SMAGs were successful in implementing the intended interventions and provided additional interventions beyond their prescribed roles; deficiencies found due to inadequate funding and support [48] |

(*Continued*)

**Table 2.** (Continued)

| Intervention type | Summary of outcomes |
|---|---|
| SMAGs were implemented to increase awareness of maternal and neonatal health services and to link the community with available services, including ANC, PNC and SBA visits [49] | Increased odds of attending at least 4 ANC visits (aOR 1.63; 95% CI 1.38–1.99); increased SBA (aOR 1.72; 95% CI 1.38–1.99) [49] |
| Community meetings/education and community motivators; renovation and supplies to primary health units; transport to district hospital via 4-wheel drive vehicle [50] | Increase in women with obstetric complications seen at primary health units; decreased referral by community motivators; cost of intervention was $5,082 [50] |
| SMGL program: Community engagement-specific interventions included training and engagement of community groups and leaders; SMAGs in some districts [52] | Provider knowledge, clinical confidence, and job satisfaction were higher in intervention districts; increased patient satisfaction in Ugandan sites for equipment availability, provider knowledge / communication skills, and care quality [52] |
| Millennium Villages Project (MVP), a multi-country rural development project spanning across multiple sectors. Community engagement-specific health interventions included integrating community case management through CHWs [56] | Impact estimates for 30 of the 40 sites were significant; no impact on consumption-based measures of poverty; decreased on-site spending from $132 per person to $109 from the first half of the project to the second half; increased proportion of births attended by a skilled birth attendant; increased ANC visits [56] |
| Developing or strengthening community-based systems to provide escort pregnant women to facilities and community-based linkages to health facilities [58] | Increased proportion facility-based deliveries in both Uganda (46% to 67%) and Zambia (63% to 90%) [58] |
| Integration of emergency transport schemes (ETS) with community-based demand-generating activities, including support groups, training TBAs and engaging religious leaders [60] | The ETS model that integrated with the community to create demand-generating activities was perceived as more effective, resulting in a reported increase in maternal and child health services [60] |
| mAccess program: an interactive and personalized 2-way SMS messaging to increase maternal and child health service utilization along with SMS messaging for identification of community-based motorcycle transport to health facilities; integration of CHWs [61] | Acceptance of the mAccess innovation throughout pregnancy and childbirth; assistance in overcoming some health system barriers [61] |
| Volunteer CHWs were trained to provide basic health information and antenatal and postnatal visits combined with the mAccess program, which provides 2-way text messaging to increase maternal and child health service utilization and identify motorcycle transport for maternal emergencies [62] | 5-times increased odds of 4 ANC visits; 3 times higher odds of travel time of 30–60 minutes to a health facility; 4 times higher odds of at least 4 PNC visits; no differences in miscarriage or stillbirth rates [62] |
| Development of women's groups; health education intervention [65] | Identification of strategies to address maternal health, neonatal, and infant health problems [65] |
| Training community maternal health workers to accompany women to maternal health visits and provide education on importance of facility-based deliveries; maternal waiting house [68] | Increase in average monthly antenatal care visits from 20 to 32; increase from 46 facility-based deliveries to 178 and 216 in the first and second years; 49 patients transported to hospital with obstetric complications [68] |
| Community capacity building and engagement to develop emergency transportation plans [69] | Increase in number of villages with community plans for maternal health emergency transportation [69] |
| SMGL program interventions targeting the first delay: communication and education to communities via media and community events, increasing partnerships between communities and facilities, community monitoring and evaluation, community activities to increase awareness of facility services and birth preparedness, increase delivery of preventative services [72] | Increased proportion of facility-based deliveries in both Uganda (45.5% to 66.8%) and Zambia (62.6% to 90.2%) [72] |
| SMGL program: Community engagement-specific interventions included training and engagement of community groups and leaders; SMAGs in some districts [73] | Increase in institutional deliveries of 47% in Uganda and 44% in Zambia; decrease in MMR in intervention facilities of 44% in Uganda and 38% in Zambia; decreased maternal death due to obstetric hemorrhage by 42% in Uganda and 65% in Zambia [73] |

(*Continued*)

**Table 2.** (Continued)

| Intervention type | Summary of outcomes |
|---|---|
| The Mozambique Community-Level Interventions for Pre-eclampsia (CLIP): community engagement and CHW assessments and visits to evaluate and manage pregnancy hypertension [74] | No decrease in maternal, fetal, or newborn mortality / morbidity [74] |
| CLIP Program: community engagement and CHW assessments and visits to evaluate and manage pregnancy hypertension [75] | Development of an approach to evaluation of a complex health intervention in a low resource setting [75] |
| Community-based volunteer groups (SMAGs) as health promoters to provide education, encourage ANC/PNC visits, facility-based deliveries, and identify complications [77] | Positive perception of the program and reported increase in maternal health services [77] |
| Multifaceted approach which included community sensitization [82] | Increased receipt of ANC components and SBA attendance at delivery, increased facility-based deliveries; no difference in coverage of 4 ANC visits or PNC visits [82] |
| **Education/Training** [23,24,27–29,33,34,36,38,40,41,46,48–54,56,60,65,68,74,75,77,82] | |
| Train the trainer curriculum for VHWs to enhance skills in community mobilization, participatory decision-making and communication [23] | Increased knowledge of danger signs, birth planning, timely referrals, and transport to hospitals; support and retention of VHWs [23] |
| Six-week long training course for VHWs in order to provide education to pregnant women and families on maternal and newborn health, birth planning, and recognition of danger signs [24] | Sustained VHW involvement in education, continued village emergency transport system use. Improved early prenatal care, identification of pregnancy danger signs, outcome data in targeted districts [24] |
| Maternal package with trained midwives including a mentoring program to provide continuing professional development and phone advice [27] | No significant effect on stillbirth or facility delivery rates. Women were positively influenced to deliver in a health facility if their husbands were involved in the decision concerning the place of birth and if they had experience in the health center [27] |
| CHWs were trained by health workers to provide education to community members through a program called Home Based Life Saving Skills which covered topics on maternal and newborn complications using cards depicting complications on one side and actions for treatment on the other side [28] | Increased facility-based deliveries after training, increased ANC visits, increased knowledge of danger signs and birth plan/complication readiness [28] |
| Training of healthcare workers [29] | Challenges with implementation included failure to consider TBAs, neglect of equity and quality of care [29] |
| SMGL program: Education and training-specific interventions included behavior change communication through mass media, drama kits and documentaries; training of healthcare providers [33] | 44% reduction in both facility and district wide MMR in Uganda, and a 38% decrease in facility and a 41% decline in districtwide MMR in Zambia [33] |
| Midwife training in safe motherhood and life saving skills; healthcare provider refresher trainings on maternal care [34] | Increases in antenatal/postnatal visits and facility-based deliveries [34] |
| SMAGs to discuss safe pregnancy, delivery, and neonatal care. Encouraged communities to ensure transport and social support were available for trips to healthcare facilities [36] | Improved women's knowledge of antenatal care and obstetric danger signs; increased use of emergency transport; increased deliveries involving a skilled birth attendant [36] |
| Health system strengthening through improvements including staff trainings [38] | Increase in the proportion of women seeking ANC of +11.4% and +10.3% in the intervention and non-intervention groups, respectively; increase the the proportion of women seeking perinatal care of +66.2% in the intervention group and +28.9% in the non-intervention group [38] |
| Staff trainings on emergency obstetric care [40] | Increase in use of referral facility by women from areas with community loans; cost of community immobilization was $472 USD [40] |

(*Continued*)

**Table 2.** (Continued)

| Intervention type | Summary of outcomes |
|---|---|
| Essential obstetric care training for staff at district health centres [41] | Increase in the number of women receiving emergency obstetric care; increased rate of major obstetric interventions; decreased risk of death for women undergoing emergency obstetric treatment 2 years after intervention; decreased maternal mortality for women referred to facilities [41] |
| SMGL program: Education and training-specific interventions included behavior change communication through mass media, drama kits and documentaries; training of healthcare providers [46] | Community volunteer groups were formalized and institutionalized; continued leadership promoting maternal health; increased maternity waiting homes; cost effectiveness via cost of death averted between $177–206 per life year gained [46] |
| SMAGs were implemented to increase awareness of maternal and neonatal health services and to link the community with available services, including ANC, PNC and SBA visits [48] | SMAGs were successful in implementing the intended interventions and provided additional interventions beyond their prescribed roles; deficiencies found due to inadequate funding and support [48] |
| SMAGs were implemented to increase awareness of maternal and neonatal health services and to link the community with available services, including ANC, PNC and SBA visits [49] | Increased odds of attending at least 4 ANC visits (aOR 1.63; 95% CI 1.38–1.99); increased SBA (aOR 1.72; 95% CI 1.38–1.99) [49] |
| Refresher training obstetric complication management for facility staff in recognition of complications for referral, management of complications and recordkeeping [50] | Increase in women with obstetric complications seen at primary health units; decreased referral by community motivators; cost of intervention was $5,082 [50] |
| Quality improvement team training on criterion-based audit and maternal death audits [51] | Improvement of 4 of 7 standards: adequate resuscitation before referral, <2 hours from time of ambulance called to hospital arrival, clinical eval within 30 min of patient arrival, and feedback to referring health center [51] |
| SMGL program: Education/training-specific interventions included obstetric provider training and mentoring by experienced clinicians [52] | Provider knowledge, clinical confidence, and job satisfaction were higher in intervention districts; increased patient satisfaction in Ugandan sites for equipment availability, provider knowledge / communication skills, and care quality [52] |
| Community education on pregnancy-related complications [53] | Increase in percentage of births in basic or comprehensive emergency obstetric facilities from 17–24%; decrease in case fatality rate among women with obstetric facilities from 9.4 to 1.9% [53] |
| Promotion of birth plans during ANC visits by health healthcare providers [54] | Women in the intervention arm were more likely to deliver in a health unit and to seek postnatal care within 1 month of delivery than women in the control arm; no difference in provider or patient satisfaction [54] |
| Millennium Villages Project (MVP), a multi-country rural development project spanning across multiple sectors. Education and training-specific health interventions included promotion of modern family planning methods and skilled birth attendance [56] | Impact estimates for 30 of the 40 sites were significant; no impact on consumption-based measures of poverty; decreased on-site spending from $132 per person to $109 from the first half of the project to the second half; increased proportion of births attended by a skilled birth attendant; increased ANC visits [56] |
| Integration of emergency transport schemes (ETS) with community-based demand-generating activities; training workshop for volunteer transporters focused on providing effective emergency transport during obstetric complications [60] | The ETS model that integrated with the community to create demand-generating activities was perceived as more effective, resulting in a reported increase in maternal and child health services [60] |
| Development of women's groups; health education intervention [65] | Identification of strategies to address maternal health, neonatal, and infant health problems [65] |

(*Continued*)

**Table 2.** (Continued)

| Intervention type | Summary of outcomes |
|---|---|
| Training community maternal health workers to accompany women to maternal health visits and provide education on importance of facility-based deliveries; maternal waiting house [68] | Increase in average monthly antenatal care visits from 20 to 32; increase from 46 facility-based deliveries to 178 and 216 in the first and second years; 49 patients transported to hospital with obstetric complications [68] |
| The Mozambique Community-Level Interventions for Pre-eclampsia (CLIP): community education on awareness of pre-eclampsia and eclampsia recognition and complications; birth preparedness, complications, savings, transportation planning [74] | No decrease in maternal, fetal, or newborn mortality / morbidity [74] |
| CLIP Program: community education on awareness of pre-eclampsia and eclampsia recognition and complications; birth preparedness, complications, savings, transportation planning [75] | Development of an approach to evaluation of a complex health intervention in a low resource setting [75] |
| Community-based volunteer groups (SMAGs) as health promoters to provide education, encourage ANC/PNC visits, facility-based deliveries, and identify complications [77] | Positive perception of the program and reported increase in maternal health services [77] |
| Multifaceted approach which included heath worker trainings [82] | Increased receipt of ANC components and SBA attendance at delivery, increased facility-based deliveries; no difference in coverage of 4 ANC visits or PNC visits [82] |
| **Financial** [26,29,30,34,35,37–41,44,55,56,59,63,72,73,82] | |
| Comprehensive voucher program covering delivery of services at public and private health facilities and round-trip transportation provided by private sector workers [26] | Increase on demand for births at health facilities, cost-effectiveness of the voucher program [26] |
| National health policy providing partial subsidies for childbirth and emergency obstetrical and neonatal care, free referral ambulance cost, medications and supplies for deliveries, training of healthcare workers [29] | Challenges with implementation included failure to consider TBAs, neglect of equity and quality of care [29] |
| Voucher that covered access to four antenatal care visits, a facility-based deliver (including treatment of complications if any, and a postnatal visit at the mother's choice of qualified health facility) [30] | Voucher program was associated with increased facility-based deliveries [30] |
| Voucher for antenatal/postnatal visits, facility-based deliveries, and locally available transport [34] | Increases in antenatal/postnatal visits and facility-based deliveries [34] |
| Support of saving groups to encourage small scale savings and provide credit [35] | Increased awareness of birth and neonatal health issues, improved access to transportation via linkage from savings groups [35] |
| Establishment of a community emergency loan fund [37] | Contribution to the loan fund and participation in the voluntary transport service by transport owners steadily increased with each subsequent launch of the program; in 9 months, 18 women were approved for loans and 18 women were transported; costs of launching and maintaining the program were estimated at $5,681 [37] |
| Health system strengthening through multiple interventions, including removal of user fees [38] | Increase in the proportion of women seeking ANC of +11.4% and +10.3% in the intervention and non-intervention groups, respectively; increase the the proportion of women seeking perinatal care of +66.2% in the intervention group and +28.9% in the non-intervention group [38] |
| "transportMYpatient" program using mobile technology and local ambassadors to provide transportation funds for referral to a specific hospital in Dar Es Salaam for obstetric fistula [39] | Increase in referred patients for obstetric fistula repair; referrals from 18 of 20 mainland regions; average cost of 1-way transport was $28, increased to $37 when cost from bus station to hospital, food, and ambassador fees were included [39] |

(*Continued*)

**Table 2.** (Continued)

| Intervention type | Summary of outcomes |
|---|---|
| Community-generated loan funds [40] | Increase in use of referral facility by women from areas with community loans; cost of community immobilization was $472 USD [40] |
| Community cost-sharing schemes [41] | Increase in the number of women receiving emergency obstetric care; increased rate of major obstetric interventions; decreased risk of death for women undergoing emergency obstetric treatment 2 years after intervention; decreased maternal mortality for women referred to facilities [41] |
| Reduced cost of antenatal care are provided by an NGO (Midwives At Maternity Azur Clinic) [44] | Reports of burial reduction from once daily to once per week [44] |
| Provision of transport vouchers for pregnancy-related visits [55] | Transport vouchers increased institutional deliveries by 42.9% and 30% for baby kits; antenatal visits increased by 60% and postnatal visits by 49.2% for vouchers; cost per delivery was $9.4 USD per transport voucher and $10.5 USD per baby kit; incremental cost per unit, increment in institutional deliveries of $15.9 USD for vouchers, and $30.6 USD for baby kit [55] |
| Millennium Villages Project (MVP), a multi-country rural development project spanning across multiple sectors. Financial-specific health interventions included combating poverty through enterprise diversification and business development, elimination of user fees [56] | Impact estimates for 30 of the 40 sites were significant; no impact on consumption-based measures of poverty; decreased on-site spending from $132 per person to $109 from the first half of the project to the second half [56] |
| Vouchers for facility-based deliveries [59] | Increased odds of facility-based delivery at voucher sites [59] |
| Transport and service vouchers for maternal health services [63] | Increased antenatal and postnatal care, increased facility-based deliveries; increased income and satisfaction from local transporters; total cost of the project at $180,406 USD [63] |
| SMGL program interventions targeting the first delay: communication and education to communities via media and community events, increasing partnerships between communities and facilities, community monitoring and evaluation, community activities to increase awareness of facility services and birth preparedness, increase delivery of preventative services [72] | Increased proportion of facility-based deliveries in both Uganda (45.5% to 66.8%) and Zambia (62.6% to 90.2%) [72] |
| SMGL program: Financial-specific interventions included marketing and distributing clean delivery kits, transportation vouchers, and promoting community-based loans to increase use of facility-based delivery [73] | Increase in institutional deliveries of 47% in Uganda and 44% in Zambia; decrease in MMR in intervention facilities of 44% in Uganda and 38% in Zambia; decreased maternal death due to obstetric hemorrhage by 42% in Uganda and 65% in Zambia [73] |
| Multifaceted approach, which included free ambulance transport, removal of facility user fees [82] | Increased receipt of ANC components and SBA attendance at delivery, increased facility-based deliveries; no difference in coverage of 4 ANC visits or PNC visits [82] |
| **Transportation** [20,24–27,29,31,33,35,37,38,41,43–47,50,52,53,56–58,61,62,64,66,67,69–71,74–76,78–82] | |
| Ambulance-based referral system [20] | Ambulance-based referral demonstrated cost savings of $24.7 per year of life saved [20] |
| Community-supported transport for maternal emergencies [24] | Sustained VHW involvement in education, continued village emergency transport system use. Improved early prenatal care, identification of pregnancy danger signs, outcome data in targeted districts [24] |
| Development of community-supported emergency transport system via village health workers [25] | 6 out of 10 villages with community-supported transportation systems continued to function after completion of the initial intervention by integrating into the village infrastructure and governance [25] |

(*Continued*)

**Table 2.** (Continued)

| Intervention type | Summary of outcomes |
|---|---|
| Comprehensive voucher program covering delivery of services at public and private health facilities and round-trip transportation provided by private sector workers [26] | Highly cost effective with incremental cost-effectiveness ratio per DALY of $302 [26] |
| Maternal package including transport for referral of obstetric patients [27] | No significant effect on stillbirth or facility delivery rates [27] |
| Free of cost referral ambulances [29] | Challenges with implementation included failure to consider TBAs, neglect of equity and quality of care [29] |
| Motorbike ambulance system supported by community contributions [31] | Ambulances were found to be acceptable and feasible and were used regularly for obstetric emergencies as well as other emergency conditions; financial contributions were variable from households and villages [31] |
| SMGL program: Transportation-specific interventions included transportation vouchers, procurement of ambulances, motorcycles and motorbikes [33] | 44% reduction in both facility and district wide MMR in Uganda, and a 38% decrease in facility and a 41% decline in districtwide MMR in Zambia [33] |
| Improving linkage with local transporters for obstetric patients [35] | Increased awareness of birth and neonatal health issues, improved access to transportation via linkage from savings groups [35] |
| Establishment of a community emergency loan fund and mobilization of transport owners to form a community transport service [37] | Contribution to the loan fund and participation in the voluntary transport service by transport owners steadily increased with each subsequent launch of the program; in 9 months, 18 women were approved for loans and 18 women were transported; costs of launching and maintaining the program were estimated at $5,681 [37] |
| Health system strengthening through multiple interventions, including creation of an ambulance network [38] | Increase in the proportion of women seeking ANC of +11.4% and +10.3% in the intervention and non-intervention groups, respectively; increase the the proportion of women seeking perinatal care of +66.2% in the intervention group and +28.9% in the non-intervention group [38] |
| Ambulance transport and communications for referrals [41] | Increase in the number of women receiving emergency obstetric care; increased rate of major obstetric interventions; decreased risk of death for women undergoing emergency obstetric treatment 2 years after intervention; decreased maternal mortality for women referred to facilities [41] |
| Free national ambulance system and upgraded mobile network [43] | Pregnancy-related mortality decreased by half in districts that had implemented the ambulance system [43] |
| Transport for ANC visits and health facilities when required [44] | Reports of burial reduction from once daily to once per week [44] |
| Free regional ambulance referral system to health facilities [45] | Increased proportion of facility births from 5.8–5.9% to 13.3% [45] |
| SMGL program: Transportation-specific interventions included transportation vouchers, procurement of ambulances, motorcycles and motorbikes [46] | Community volunteer groups were formalized and institutionalized; continued leadership promoting maternal health; increased maternity waiting homes; cost effectiveness via cost of death averted between $177–206 per life year gained [46] |
| Three motorcycle ambulances placed at remote rural health centers [47] | Decreased referral time of 35–76%; significantly cheaper costs compared to a car ambulance [47] |
| Transport to district hospital via 4-wheel drive vehicle [50] | Increase in women with obstetric complications seen at primary health units; decreased referral by community motivators; cost of intervention was $5,082 [50] |

(*Continued*)

**Table 2.** (Continued)

| Intervention type | Summary of outcomes |
|---|---|
| SMGL program: Transportation-specific interventions included transportation vouchers, procurement of ambulances, motorcycles and motorbikes [52] | Provider knowledge, clinical confidence, and job satisfaction were higher in intervention districts; increased patient satisfaction in Ugandan sites for equipment availability, provider knowledge / communication skills, and care quality [52] |
| Provision of 2 district ambulances [53] | Increase in percentage of births in basic or comprehensive emergency obstetric facilities from 17–24%; decrease in case fatality rate among women with obstetric facilities from 9.4 to 1.9% [53] |
| Millennium Villages Project (MVP), a multi-country rural development project spanning across multiple sectors. Transportation-specific interventions included strengthening of an ambulance referral system [56] | Impact estimates for 30 of the 40 sites were significant; no impact on consumption-based measures of poverty; decreased on-site spending from $132 per person to $109 from the first half of the project to the second half; increased proportion of births attended by a skilled birth attendant; increased ANC visits [56] |
| Free district ambulance referral system [57] | Increase in c-section rate from 0.57% to 1.21%; 50% increase in hospital deliveries; non-significant reduction in hospital stillbirths [57] |
| SMGL was assessed specifically targeting the ability to reach care, including decreasing the distance to facilities capable of managing emergency obstetric and newborn complications and addressing transportation challenges [58] | Access to facilities improved through integrated transportation and communication services efforts. In Uganda there was a 258% increase in facility deliveries supported by transportation vouchers. In Zambia, there was a 31% increase in health facilities with available transportation [58] |
| mAccess program: an interactive and personalized 2-way SMS messaging to increase maternal and child health service utilization along with SMS messaging for identification of motorcycle transport to health facilities [61] | Acceptance of the mAccess innovation throughout pregnancy and childbirth; assistance in overcoming some health system barriers [61] |
| mAccess program: Provision of 2-way text messaging to increase maternal and child health service utilization and identify motorcycle transport for maternal emergencies [62] | 5-times increased odds of 4 ANC visits; 3 times higher odds of travel time of 30–60 minutes to a health facility; 4 times higher odds of at least 4 PNC visits; no differences in miscarriage or stillbirth rates [62] |
| SMGL program: Transportation-specific interventions included development and strengthening of systems for communication, referral, and transportation available 24/7 [64] | 44% reduction in both facility and district wide MMR in Uganda, and a 38% decrease in facility and a 41% decline in districtwide MMR in Zambia [64] |
| Free transportation and installation of radio system to link hospitals and referral vehicles [66] | Increased hospital utilization for maternal complications; decrease in case fatality rate by half; cost of intervention was $74,836 USD [66] |
| Referral system development with ambulances and radios; policy and advocacy efforts [67] | Increased proportion of facility births; no significant change in institutional births; increase facility use for complications; case fatality rate decreased from 2.9% to 1.6% [67] |
| Community capacity building and engagement to develop emergency transportation plans [69] | Increase in number of villages with community plans for maternal health emergency transportation [69] |
| SMGL program: Transportation-specific interventions included expansion of motorized transportation to emergency obstetric and newborn care (EmONC) through transportation vouchers for motorcycle taxis and a coordinated ambulance service [70] | Decrease in estimated travel time to EmONC facilities; increase in proportion of EmONC and comprehensive EmONC (CEmONC) facilities within 2 hours by 18% and 37%, respectively, from baseline; further increases in EmONC and CEmONC facility use occurred if 4-wheeled vehicles could be used (14% and 31%) [70] |
| Issue of 18 dedicated vehicles for interfacility transport for maternal health emergencies [71] | Decrease in maternal mortality rate from 279/100,000 to 152/100,000; decrease in dispatch interval for referrals from 32.01 to 22.47 minutes; increase in number of vehicles dispatched within 1 hour [71] |

(*Continued*)

**Table 2.** (Continued)

| Intervention type | Summary of outcomes |
|---|---|
| CLIP program: community transport scheme was designed to ensure adequate emergency obstetric care-preparedness among pregnant women and communities [74] | No decrease in maternal, fetal, or newborn mortality / morbidity [74] |
| CLIP Program: community transport scheme was designed to ensure adequate emergency obstetric care-preparedness among pregnant women and communities [75] | Development of an approach to evaluation of a complex health intervention in a low resource setting [75] |
| Training and cooperation of local transport workers union to develop a community transport system for maternal health emergencies [76] | Mean cost of transport $5.89 USD, $268 for cost of full transport intervention; 29 transports for obstetric complications [76] |
| Free hospital-based referral ambulance [78] | Hospital referral ambulance is cost effective; cost savings of $15.82 USD/year; cost-effectiveness holds to a critical referral rate of 1.3% (1 critical referral for 77 "useless" referrals [78] |
| Establishment of a Méèdecins Sans Frontières-run central emergency obstetric reference center and implementation of an effective communication and transport system to ensure access [79] | Estimated 55 deaths (74%) in the district were averted; MMR of 208/100,000 [79] |
| Implementation of an effective communication and transport system with 3 ambulances for emergency obstetric care [80] | Median ambulance referral time of 78 minutes; cost of €61/obstetric case transferred; increased risk of death associated with referral times >3 hours; 80% coverage of complicated obstetric cases and 92% coverage for c-sections [80] |
| Free hospital-based ambulance referral system [81] | Average of 2.9 referrals/day; 97.2% of referrals were for pregnancy-related complication; no maternal deaths recorded; cost of $18.47 USD per patient referred [81] |
| Multifaceted approach, which included referral strengthening and free ambulance transport [82] | Increased receipt of ANC components and SBA attendance at delivery, increased facility-based deliveries; no difference in coverage of 4 ANC visits or PNC visits [82] |
| **Facility Infrastructure** [27,33,34,38,41,44,46,50,52,53,56,58,64,67,70,82] | |
| Maternal package with trained midwives with a mentoring program, transport for referral, and equipment and accommodation for the midwives, equipment and beds for health facilities [27] | No significant effect on stillbirth or facility delivery rates [27] |
| SMGL program: Facility infrastructure interventions included improved quality of facilities, provision of essential medicines and medical commodities and hiring of healthcare personnel [33] | 44% reduction in both facility and district wide MMR in Uganda, and a 38% decrease in facility and a 41% decline in districtwide MMR in Zambia [33] |
| Healthcare provider training and small updates to medications and supplies [34] | Increases in antenatal/postnatal visits and facility-based deliveries [34] |
| Health system strengthening including improvements in facility structure and equipment [38] | Increase in the proportion of women seeking ANC of +11.4% and +10.3% in the intervention and non-intervention groups, respectively; increase the the proportion of women seeking perinatal care of +66.2% in the intervention group and +28.9% in the non-intervention group [38] |
| Equipment for emergency obstetric management [41] | Increase in the number of women receiving emergency obstetric care; increased rate of major obstetric interventions; decreased risk of death for women undergoing emergency obstetric treatment 2 years after intervention; decreased maternal mortality for women referred to facilities [41] |
| NGO-supported clinic (Midwives At Maternity Azur Clinic); maternal waiting home for high risk patients [44] | Reports of burial reduction from once daily to once per week [44] |

(*Continued*)

**Table 2.** (Continued)

| Intervention type | Summary of outcomes |
|---|---|
| Facility infrastructure interventions included improved quality of facilities, provision of essential medicines and medical commodities and hiring of healthcare personnel [46] | Increased efficiency in allocation of resources for maternal and newborn health; improved management capacities; community volunteer groups were formalized and institutionalized; increased maternity waiting homes; cost effectiveness via cost of death averted between $177–206 per life year gained [46] |
| Renovation and supplies to primary health units [50] | Increase in women with obstetric complications seen at primary health units; decreased referral by community motivators; cost of intervention was $5,082 [50] |
| SMGL program: Facility infrastructure-specific interventions included provision of new equipment and supplies; upgrades of maternity wards; and new operating theaters in Uganda [52] | Provider knowledge, clinical confidence, and job satisfaction were higher in intervention districts; increased patient satisfaction in Ugandan sites for equipment availability, provider knowledge / communication skills, and care quality [52] |
| Increasing skilled staff at facilities; upgrading facilities [53] | Increase in percentage of births in basic or comprehensive emergency obstetric facilities from 17–24%; decrease in case fatality rate among women with obstetric facilities from 9.4 to 1.9% [53] |
| Millennium Villages Project (MVP), a multi-country rural development project spanning across multiple sectors. Facility infrastructure-specific health interventions included construction of health facilities and staff housing, introduction of electronic health record systems [56] | Impact estimates for 30 of the 40 sites were significant; no impact on consumption-based measures of poverty; decreased on-site spending from $132 per person to $109 from the first half of the project to the second half; increased proportion of births attended by a skilled birth attendant; increased ANC visits [56] |
| SMGL was assessed specifically targeting the ability to reach care, including increasing the number of EmONC facilities [58] | Access to facilities improved through integrated transportation and communication services efforts. In Uganda there was a 258% increase in facility deliveries supported by transportation vouchers. In Zambia,there was a 31% increase in health facilities with available transportation. Renovation and construction of maternity waiting homes resulted in a 69% increase in facilities associated with maternity waiting homes [58] |
| SMGL program: Facility infrastructure-specific interventions included improvement of safe facilities and hospitals for deliveries; supplies and provision of basic and emergency obstetric services [64] | 44% reduction in both facility and district wide MMR in Uganda, and a 38% decrease in facility and a 41% decline in districtwide MMR in Zambia [64] |
| Facility infrastructure improvements [67] | Increased proportion of facility births; no significant change in institutional births; increase facility use for complications; case fatality rate decreased from 2.9% to 1.6% [67] |
| SMGL program: Facility infrastructure-specific interventions included improving the availability and distribution of EmONC services [70] | Decrease in estimated travel time to EmONC facilities; increase in proportion of EmONC and comprehensive CEmONC facilities within 2 hours by 18% and 37%, respectively, from baseline; further increases in EmONC and CEmONC facility use occurred if 4-wheeled vehicles could be used (14% and 31%) [70] |
| A multifaceted approach, which included health information system strengthening, improvement of health center infrastructure and supplies [82] | Increased facility-based deliveries from 7.3% to 35.6%; increased receipt of ANC components and SBA attendance at delivery; no difference in coverage of 4 ANC visits or PNC visits [82] |

review, no articles were included that utilized facility infrastructure improvements as the sole intervention; rather all such interventions were complementary to other engagement, education, financing, and transportation programs. Multiple studies evaluated different aspects of the same program or intervention.

The majority of community-based interventions (24 out of 28) involved some aspect of training and integrating community members such as village health workers or community health teams. Ten of the 27 interventions in the education/training category focused on health-care worker education and training while the remainder focused on education to lay-people and community members. Thirteen of the financial interventions involved the use of a voucher program for maternal health services and/or transport or removal of user fees; 5 involved the development of a community loan program or cost-sharing scheme. The remaining financial interventions included the use of mobile technology to directly provide funds once patients requiring referral were identified, and a national policy change to subsidize certain aspects of maternal care, including transportation. Twenty-four transportation-focused interventions involved the development of an ambulance-based referral system or ambulance network. Eleven of the interventions focused on community supported transportation systems for obstetric emergencies and 4 involved the use of vouchers for transportation services. Interventions included the creative use of several different types of transportation vehicles (ambulances, 4-wheel drive vehicles, motorbikes, bicycles, donkey carts and boats). Finally, there were 16 interventions that included some aspect of improving facility infrastructure or providing equipment for healthcare facilities.

## Discussion

This review identified a variety of obstetric access to care and transportation interventions across several African nations that could be translated to other emergency conditions. Though cases and sample sizes varied greatly from a few dozen to the thousands, several themes emerged. Many interventions utilized a "local arbiter", midwife or otherwise, to identify cases needing interventional support. In some instances, support was in the form of financial loans or direct payments for transportation, vouchers for public transport, or existing ambulance utilization. There is a noted significant heterogeneity in the outcome measurements, from mortality measures, to the number of individuals educated, to the financial indicators of money raised/disbursed. Although high quality evidence is lacking, many of these interventions could be adapted to address other emergent conditions requiring transportation to access care.

A previous systematic review evaluating access barriers to obstetric care in sub-Saharan Africa identified multiple demand-side barriers including household resources or income, lack of transport availability, cost of transportation, poor information on health care services and providers, stigma and self-esteem concerns, lack of birth preparation, and cultural beliefs [83]. Wilson et al. conducted a systematic review specifically evaluating qualitative studies on maternal emergency transport in LMICs and found key issues to be the availability and speed of transport, terrain, weather, support, lack of autonomy in decision-making for women, cultural barriers, cost, and ergonomics during transport [84]. These findings are reflected in the emergency care literature as well. A more recent systematic review evaluated more broadly the barriers to out-of-hospital emergency care in low and middle-income countries, and found many of the same barriers as in the obstetric literature, including cultural and transportation barriers [85]. The interventions described in our systematic review focus specifically on access-to-care and transportation for obstetric conditions, including emergency obstetrics. We found that the interventions identified in this systematic review address many of the barriers also found more broadly in emergency care that have not been fully described in the obstetrics literature, such as communication and coordination, training of personnel, and infrastructure [85].

This review highlights broad diversity in the approaches to access-to-care interventions for obstetric care. Although rural and urban contexts have divergent needs for improving access to obstetric care, the assorted approaches taking place across the continent of Africa have varied strengths and benefits. Our study also highlights gaps in knowledge and expertise in this subset of patients. Studies were predominantly clustered in eastern and sub-Saharan Africa. Numerous countries had no representative published literature within the scope of this review and many had only one such article. In cases where the single study from a country may have a small population size, Fig 1 may visually mislead the extent of geographic spread of the reviewed literature. These findings resonate with previous literature describing the presence of EMS systems in eastern Africa, with a relative dearth of established EMS systems in West Africa [1].

Prior work by Ehiri et al. in 2018 does have some overlap, though their study was more narrowly focused on ameliorating adverse birth outcomes for both mothers and babies and focused solely on transportation solutions [86]. Our systematic review looked more comprehensively at other solutions in access-to-care, such as training and educational interventions and community engagement, more closely mirroring the three-delays model. Furthermore, their work looked at LMICs more broadly, while our review was more focused in geographic scope.

The majority of the studies included in our qualitative synthesis were classified as having a high degree of bias, indicating concerns around the quality of studies on this subject in this geographic area. There was only one randomized-control trial and most of the studies were observational or cross-sectional. This follows findings in the global emergency medicine literature, where there has been a paucity of high quality literature focusing on interventions to increase access to emergency care in LMICs [87,88]. This spurs us to creatively evaluate and assess other types of interventions that can be translated or incorporated into efforts to increase access to emergency care. Valuable lessons from literature around obstetric care can be applied towards addressing gaps in access to care for other emergent conditions. Many recent advances in emergency obstetric care are the result of a multifactorial approach to achieving appropriate care using the three-delays model as a framework [15]. This conceptual framework has been proposed as a viable model by which to approach the delivery of and access to emergency care [16].

Many of the interventions identified in this systematic review are centered around the first and second delays in the conceptual model (delays in deciding to seek care and delays in the travel/transport required to obtain care). Previous literature describing barriers to out-of-hospital emergency care have identified specific obstacles that fit within the first two delays: decision to call EMS, recognition of emergency health conditions, challenges in finding transportation, delays in transportation, and access to appropriate hospitals [85]. One must also consider that even if a patient is transported to an appropriate hospital, there may be a paucity of blood products, medications, or adequate surgical service availability. We found that several of the interventions identified in this review could be translated to the emergency care for a number of different conditions. For example, the provision of vouchers for transportation, community-supported ambulances or transportation schemes, community cost-sharing schemes, and community education are all approaches that could be applied to improving access to numerous emergency conditions and addressing known barriers.

## Limitations

This review was initially meant to encompass interventions addressing both trauma and emergency obstetric interventions in Africa. However, given the paucity of trauma-specific

interventions and the relative wealth of studies focusing on emergency obstetric interventions, the focus of the review was pivoted to translating emergency obstetric interventions to broader emergency care interventions. We also chose to focus only on interventions in Africa, thus we may have overlooked pertinent interventions and better quality data in other LMIC settings.

## Conclusion

Interventions to improve access to emergency obstetric conditions in Africa are varied in quality and focus. Many of these interventions follow the three-delays conceptual model and can thus be translated to improving access to a broader array of emergent conditions. In particular, interventions focusing on improving the first two delays in care through education, community involvement, and improving transportation to healthcare facilities, hold the most promise for future efforts at improving emergency access to care for both obstetric as well as other conditions.

## Supporting information

**S1 Table. PRISMA checklist.**
(DOCX)

**S2 Table. Search terms.**
(DOCX)

**S3 Table. Quality assessment.**
(XLSX)

## Author Contributions

**Conceptualization:** Anjni Joiner, Francis Sakita, Megan von Isenburg, Catherine Staton, Joao Ricardo Nickenig Vissoci.

**Data curation:** Anjni Joiner, Austin Lee, Phindile Chowa, Ramu Kharel, Lekshmi Kumar, Nayara Malheiros Caruzzo, Thais Ramirez, Lindy Reynolds, Lee Van Vleet, Megan von Isenburg, Anna Quay Yaffee, Joao Ricardo Nickenig Vissoci.

**Formal analysis:** Anjni Joiner, Austin Lee, Phindile Chowa, Ramu Kharel, Lekshmi Kumar, Nayara Malheiros Caruzzo, Thais Ramirez, Lindy Reynolds, Lee Van Vleet, Anna Quay Yaffee.

**Funding acquisition:** Anjni Joiner.

**Methodology:** Anjni Joiner, Megan von Isenburg, Catherine Staton, Joao Ricardo Nickenig Vissoci.

**Project administration:** Anjni Joiner.

**Resources:** Megan von Isenburg, Joao Ricardo Nickenig Vissoci.

**Software:** Megan von Isenburg.

**Supervision:** Anjni Joiner, Francis Sakita, Catherine Staton, Joao Ricardo Nickenig Vissoci.

**Validation:** Thais Ramirez.

**Visualization:** Anjni Joiner, Austin Lee, Nayara Malheiros Caruzzo, Thais Ramirez, Joao Ricardo Nickenig Vissoci.

**Writing – original draft:** Anjni Joiner, Austin Lee, Nayara Malheiros Caruzzo, Thais Ramirez, Francis Sakita, Megan von Isenburg, Joao Ricardo Nickenig Vissoci.

**Writing – review & editing:** Anjni Joiner, Austin Lee, Phindile Chowa, Ramu Kharel, Lekshmi Kumar, Nayara Malheiros Caruzzo, Thais Ramirez, Lindy Reynolds, Francis Sakita, Lee Van Vleet, Megan von Isenburg, Anna Quay Yaffee, Catherine Staton, Joao Ricardo Nickenig Vissoci.

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
