## [Decision Letter · Decision Letter 0]

21 Apr 2021

PONE-D-20-31712

Access to Care Solutions in Healthcare for Obstetric Care in Africa: A Systematic Review

PLOS ONE

Dear Dr. Anjni Joiner,

Thank you for submitting your manuscript to PLOS ONE. After careful consideration, we feel that it has merit but does not fully meet PLOS ONE’s publication criteria as it currently stands. Therefore, we invite you to submit a revised version of the manuscript that addresses the points raised during the review process.

We look forward to receiving your revised manuscript.

Kind regards,

Limakatso Lebina, MBChB

Academic Editor

PLOS ONE

Journal Requirements:

We note you have included a table to which you do not refer in the text of your manuscript. Please ensure that you refer to Table 2 in your text; if accepted, production will need this reference to link the reader to the Table.

We note that Figure 2 in your submission contain map images which may be copyrighted. All PLOS content is published under the Creative Commons Attribution License (CC BY 4.0), which means that the manuscript, images, and Supporting Information files will be freely available online, and any third party is permitted to access, download, copy, distribute, and use these materials in any way, even commercially, with proper attribution. For these reasons, we cannot publish previously copyrighted maps or satellite images created using proprietary data, such as Google software (Google Maps, Street View, and Earth). For more information, see our copyright guidelines: http://journals.plos.org/plosone/s/licenses-and-copyright.

3a,  You may seek permission from the original copyright holder of Figure 2 to publish the content specifically under the CC BY 4.0 license. 

3b,  If you are unable to obtain permission from the original copyright holder to publish these figures under the CC BY 4.0 license or if the copyright holder’s requirements are incompatible with the CC BY 4.0 license, please either i) remove the figure or ii) supply a replacement figure that complies with the CC BY 4.0 license. Please check copyright information on all replacement figures and update the figure caption with source information. If applicable, please specify in the figure caption text when a figure is similar but not identical to the original image and is therefore for illustrative purposes only.

Please include captions for your Supporting Information files at the end of your manuscript, and update any in-text citations to match accordingly. Please see our Supporting Information guidelines for more information: http://journals.plos.org/plosone/s/supporting-information.

Please amend either the abstract on the online submission form (via Edit Submission) or the abstract in the manuscript so that they are identical.

Reviewers' comments:

Reviewer's Responses to Questions

**Comments to the Author**

1. Is the manuscript technically sound, and do the data support the conclusions?

Reviewer #1: Yes

Reviewer #2: Partly

2. Has the statistical analysis been performed appropriately and rigorously? 

Reviewer #1: N/A

Reviewer #2: N/A

3. Have the authors made all data underlying the findings in their manuscript fully available?

Reviewer #1: Yes

Reviewer #2: Yes

4. Is the manuscript presented in an intelligible fashion and written in standard English?

Reviewer #1: Yes

Reviewer #2: Yes

5. Review Comments to the Author

Reviewer #1: This is an important piece of work. The paper seems like a useful addition to the issue of emergency obstetric care in Africa. It is well written and seems to have gone through a sound systematic review process.

The very large table with all of the details of the study on it is quite hard to read - but I am not sure if there is a better way to present that data.

Additionally in my experience in Africa, even when a woman makes the decision to seek help and barriers are removed for her to do so - she may not get to a health centre with blood/a trained and available medical officer etc. You present a much more optimistic view and I hope this is indeed the case.

The problems within the health system in rural Uganda for example were very difficult to overcome - when combined with limited finances and large distances between staffed and equipped health centres.

While the conclusion is reasonable I wonder if it would be good to add something about the reality of so many of the health centres in rural Africa - that even if a woman is able to get to a facility with a trained health professional in a timely fashion (make the decision and get the transport there) there is a choice to be made about which health facility (private, Health Centre II, III, IV, V for Uganda for example) - and even then there is no guarantee that blood or a C-section will be available in the time frame needed. There is no simple solution to obstetric emergencies in Africa, but the review does provide some insights into things that are a good start.

Reviewer #2: Thank you for the opportunity to review the manuscript "Access to Care Solutions in Healthcare for Obstetric Care in Africa: A systematic review." This is an important topic, as obstetric care is a critical intervention for reducing maternal mortality, and increasing access to such services, particularly in developing countries, can dramatically reduce deaths from pregnancy complications. The authors provide a useful way to categorize the types of EMS systems in the context of obstetric care in Africa, and capture the wide diversity of intervention that have been put in place. A few points that would benefit from clarification, as follows:

• Clarify the definition Emergency Medical Services (EMS) systems in the context of obstetric care – are these only referring to transportation systems, or would it include task shifting initiatives, as well as upgrades to clinics to enable treatment of obstetric emergencies (e.g., this initiative in Tanzania that does both - https://healthmarketinnovations.org/program/bloomberg-philanthropies-maternal-health-initiative)

• Related to the above, the authors should clarify the inclusion and exclusion criteria – in particular, it wasn’t clear to me why in-hospital solutions intended to address lack of access to services excluded? (also the results section suggests otherwise, e.g., the interventions that aim to improve ‘facility infrastructure’ to me seem in part ‘in hospital solutions’)

• The discussion section states that the systematic review was more comprehensive than Ehiri et al. 2018 and followed “more closely mirroring the three delays model”- it would be useful to introduce the concept of the ‘three delays model’ earlier on in the paper and provide more background on this so as to better frame the results and discussion

• The authors assess the risk of bias – it would be good for the authors to provide a high-level explanation of how these assessments are done (i.e., what constitutes high /low risk) for qualitative studies as some readers (including myself) are not familiar with the Joanna Brigs checklist-

• For the data analysis, both qualitative and descriptive approaches are used- it would be good for the authors to provide a high-level explanation of the difference between qualitative and descriptive approaches for readers that may not be familiar with the nuances

6. PLOS authors have the option to publish the peer review history of their article (what does this mean?). If published, this will include your full peer review and any attached files.

Reviewer #1: No

Reviewer #2: **Yes: **Tewodaj Mengistu

---

## [Author Response · Author response to Decision Letter 0]

5 May 2021

May 5, 2021

Dear PLOS One Reviewers,

Thank you for your careful review of our submission entitled, “Access to Care Solutions in Healthcare for Obstetric Care in Africa: A Systematic Review” (PONE-D-20-31712).

We have carefully reviewed the comments and have responded as follows:

Journal Requirements:

We have carefully reviewed PLOS ONE’s style requirements, including those for file naming, and have updated the following files to conform to the requirements:

Manuscript

Revised Manuscript with Track Changes

Fig 1

Fig 2

S1 Table

S2 Table

S3 Table

2. We note you have included a table to which you do not refer in the text of your manuscript. Please ensure that you refer to Table 2 in your text; if accepted, production will need this reference to link the reader to the Table.

Thank you for noting this omittance. Table 2 is now referred to in line 205.

3. We note that Figure 2 in your submission contain map images which may be copyrighted. 

Figure 2 was elaborated by the authors using QGIS version 3.12.0, an open-source cross-platform desktop geographic information system application that supports viewing, editing, and analysis of geospatial data. This figure is not copyrighted and was instead generated by the authors. 

4. Please include captions for your Supporting Information files at the end of your manuscript, and update any in-text citations to match accordingly.

Captions for Supporting Information files were included at the end of the manuscript as outlined here:

S1 Table: PRISMA checklist

S2 Table: Search strategy

S3 Table: Quality assessment

In-text citations were updated as follows:

Supplement 1: line 108

Supplement 2: line 128

Thank you for noting this discrepancy. This has been corrected so that both abstracts are identical.

Response: The reference list was carefully reviewed to ensure completeness and to ensure that no citations were included that were retracted. No changes were made to the reference list.

Reviewer #1: 

This is an important piece of work. The paper seems like a useful addition to the issue of emergency obstetric care in Africa. It is well written and seems to have gone through a sound systematic review process.

The very large table with all of the details of the study on it is quite hard to read - but I am not sure if there is a better way to present that data.

Response: Thank you for this comment. We agree that the table is very challenging to read. However, given the heterogeneity of the studies included in this systematic review, our quality assessment necessitated multiple different assessment tools with differing data elements. We found that the using different tabs for the different types of studies was the most efficient way to represent the quality assessment.

Additionally in my experience in Africa, even when a woman makes the decision to seek help and barriers are removed for her to do so - she may not get to a health centre with blood/a trained and available medical officer etc. You present a much more optimistic view and I hope this is indeed the case.

Response: Thank you, we certainly hope that all mothers and babies can obtain adequate and appropriate care. We have additionally added a sentence in the discussion (lines 301-303) highlighting the important point of potential service/treatment unavailability.

The problems within the health system in rural Uganda for example were very difficult to overcome - when combined with limited finances and large distances between staffed and equipped health centres.

Response: Both in rural Uganda, and across Africa challenges of distance, resources, and staffing are all every day realities. Even in the rural parts of North America and Europe these challenges can lead to worsened health outcomes for some patients. Our hope is that this systematic review can highlight the wide variability in access to and quality of obstetric care in Africa, and in turn spur future research as well as targeted policy approaches and improved public health systems.

While the conclusion is reasonable I wonder if it would be good to add something about the reality of so many of the health centres in rural Africa - that even if a woman is able to get to a facility with a trained health professional in a timely fashion (make the decision and get the transport there) there is a choice to be made about which health facility (private, Health Centre II, III, IV, V for Uganda for example) - and even then there is no guarantee that blood or a C-section will be available in the time frame needed. There is no simple solution to obstetric emergencies in Africa, but the review does provide some insights into things that are a good start.

Response: We agree; as noted above, we have added a sentence in the discussion (lines 301-303) to address the potential gaps in availability of needed resources. 

Reviewer #2: 

Thank you for the opportunity to review the manuscript "Access to Care Solutions in Healthcare for Obstetric Care in Africa: A systematic review." This is an important topic, as obstetric care is a critical intervention for reducing maternal mortality, and increasing access to such services, particularly in developing countries, can dramatically reduce deaths from pregnancy complications. The authors provide a useful way to categorize the types of EMS systems in the context of obstetric care in Africa, and capture the wide diversity of intervention that have been put in place. A few points that would benefit from clarification, as follows:

• Clarify the definition Emergency Medical Services (EMS) systems in the context of obstetric care – are these only referring to transportation systems, or would it include task shifting initiatives, as well as upgrades to clinics to enable treatment of obstetric emergencies (e.g., this initiative in Tanzania that does both - https://healthmarketinnovations.org/program/bloomberg-philanthropies-maternal-health-initiative)

Response: Thank you for pointing out this valuable work in Tanzania; we agree that this type of approach is incredibly valuable and efforts to upgrade health centers as well as improve training and task shifting can be life saving. Generally, emergency medical services (EMS) are seen to be ambulance or similar services that allow for out of hospital early treatment, stabilization and transportation of patients. For the purposes of this systematic review, we were focused on interventions that looked directly at transportation or access to care solutions for prehospital or follow up care for maternal and obstetric conditions in African countries. While other efforts are certainly complementary and improved, high-quality facility-based solutions are a necessary part of the care continuum, but beyond the scope of EMS systems and our review.

• Related to the above, the authors should clarify the inclusion and exclusion criteria – in particular, it wasn’t clear to me why in-hospital solutions intended to address lack of access to services excluded? (also the results section suggests otherwise, e.g., the interventions that aim to improve ‘facility infrastructure’ to me seem in part ‘in hospital solutions’)

Response: Thank you for noting this. None of the articles included in the review were selected only due to facility infrastructure improvements, but rather, were publications highlighting multi-faceted interventions that targeted both transportation interventions amidst other approaches. As such, we did identify the theme of facility infrastructure improvements from among the sixteen articles that included such improvements, yet, none of those articles were included only because of these infrastructure approaches. As mentioned above, complementary efforts between the prehospital and in-facility care are important to enhance care across the continuum. We have added a sentence in the discussion to further clarify (lines 301-303). 

• The discussion section states that the systematic review was more comprehensive than Ehiri et al. 2018 and followed “more closely mirroring the three delays model”- it would be useful to introduce the concept of the ‘three delays model’ earlier on in the paper and provide more background on this so as to better frame the results and discussion

Response: Thank you for this comment. The three-delays model was previously introduced in the third paragraph of the introduction (although there was a discrepancy in hyphenating the term three delays, which has been corrected). The three-delays model is highlighted in the text in both the introduction (lines 91-94) and the discussion (lines 277-279 and lines 289-292).

• The authors assess the risk of bias – it would be good for the authors to provide a high-level explanation of how these assessments are done (i.e., what constitutes high /low risk) for qualitative studies as some readers (including myself) are not familiar with the Joanna Brigs checklist-

Response: The Joanna Briggs Critical Appraisal Checklist for Qualitative Research is a 10-item checklist including items such as “Congruity between the stated philosophical perspective and the research methodology” and “Representation of participants and their voices”. Responses are categorized as “yes”, “no”, “unclear” or “not applicable”. The sentence in which the Joanna Briggs checklist is introduced was revised to provide further clarification (lines 141-143).

• For the data analysis, both qualitative and descriptive approaches are used- it would be good for the authors to provide a high-level explanation of the difference between qualitative and descriptive approaches for readers that may not be familiar with the nuances

Response: Thank you for this recommendation. An additional sentence describing the difference between qualitative and descriptive approaches as pertinent to this study was added (lines 165-168).

Thank you for the opportunity to revise and strengthen our manuscript.

Sincerely,

Anjni Joiner, DO, MPH

Assistant Professor of Surgery

Division of Emergency Medicine

Duke University Medical Center

Duke Global Health Institute

US Cell Phone: (919) 413-6525

Email: anjni.joiner@duke.edu

---

## [Editor Report · Decision Letter 1]

19 May 2021

Access to Care Solutions in Healthcare for Obstetric Care in Africa: A Systematic Review

PONE-D-20-31712R1

Dear Dr. Anjni Joiner,

We’re pleased to inform you that your manuscript has been judged scientifically suitable for publication and will be formally accepted for publication once it meets all outstanding technical requirements.

Kind regards,

Limakatso Lebina, MBChB

Academic Editor

PLOS ONE

Additional Editor Comments (optional):

You have been able to adequately address all the comments and your manuscript will be accepted for publication.
---

## [Editor Report · Acceptance letter]

24 May 2021

PONE-D-20-31712R1 

Access to care solutions in healthcare for obstetric care in Africa: A systematic review 

Dear Dr. Joiner:

I'm pleased to inform you that your manuscript has been deemed suitable for publication in PLOS ONE. Congratulations! Your manuscript is now with our production department. 

Kind regards, 

on behalf of

Dr. Limakatso Lebina 

Academic Editor

PLOS ONE